# Pelvic Sidewall Anatomy in Gynecologic Oncology—New Insights into a Potential Avascular Space

**DOI:** 10.3390/diagnostics12020519

**Published:** 2022-02-17

**Authors:** Stoyan Kostov, Ilker Selçuk, Rafał Watrowski, Yavor Kornovski, Hakan Yalçın, Stanislav Slavchev, Yonka Ivanova, Deyan Dzhenkov, Angel Yordanov

**Affiliations:** 1Department of Gynecology, University Hospital “Saint Anna”, Medical University—“Prof. Dr. Paraskev Stoyanov”, 9002 Varna, Bulgaria; drstoqn.kostov@gmail.com (S.K.); ykornovski@abv.bg (Y.K.); st_slavchev@abv.bg (S.S.); yonka.ivanova@abv.bg (Y.I.); 2Department of Gynecologic Oncology, Ankara City Hospital, Woman’s Health Training and Research Hospital, University of Health Sciences, Ankara 06800, Turkey; ilkerselcukmd@hotmail.com (I.S.); drhyalcin@yahoo.com (H.Y.); 3Faculty of Medicine, University of Freiburg, 79106 Freiburg, Germany; rafal.watrowski@gmx.at; 4Department of General and Clinical Pathology, Forensic Medicine and Deontology, Division of General and Clinical Pathology, Faculty of Medicine, Medical University—“Prof. Dr. Paraskev Stoyanov”, 9002 Varna, Bulgaria; ddzhenkov@gmail.bg; 5Department of Gynecologic Oncology, Medical University Pleven, 5800 Pleven, Bulgaria

**Keywords:** pelvic sidewall anatomy, new term, new boundaries, potential avascular space, gynecologic oncology

## Abstract

The surgical treatment of gynecological malignancies is, except for tumors diagnosed at the earliest stages and patients’ desire for fertility preservation, not limited to only the affected organ. In cases of metastatic iliac lymph nodes, gynecological tumors or recurrences located near the pelvic sidewall, oncogynecologists should dissect tissues in that region. Moreover, surgery of deep infiltrating endometriosis, e.g., within the sacral plexus, or oncological procedures, such as a laterally extended endoplevic resection or a laterally extended parametrectomy, often require a dissection of the pelvic sidewall. Dissection should be meticulous, and detailed knowledge of anatomy is mandatory. There are many controversies among authors regarding the terminology in the pelvic sidewall. In particular, several imprecise or confusing definitions exist in regard to the region located medially to the psoas major muscle. Therefore, after discussing the anatomy of the pelvic sidewall and the commonly used terminology, we define a new term and boundaries of a potential avascular space, the medial psoas space. Contrary to the variety of earlier definitions, the proposed boundaries relate to a truly avascular space and could help surgeons to avoid complications resulting from misleading anatomical descriptions. Additionally, describing the clear boundaries of and possible anatomical variations in the medial psoas space may urge oncogynecologists to consider different approaches during surgery. The purpose of the present study is to describe the anatomy of the pelvic sidewall and the applications of the medial psoas space in gynecologic oncology.

## 1. Introduction

The surgical treatment of gynecological malignancies is, except for tumors diagnosed at the earliest stages and patients’ desire for fertility preservation, not limited to only the affected organ [1]. The lymphatic system is the main dissemination pathway for gynecological neoplasms. Depending on the stage and histology of a tumor, a lymph node dissection is often performed in gynecologic oncology [2]. In cases of bulky metastatic lymph nodes, gynecological tumors or recurrences located near the pelvic sidewall, oncogynecologists should dissect tissues in that region [3,4]. Moreover, surgery of deep infiltrating endometriosis, e.g., within the sacral plexus, or oncological procedures, such as a laterally extended endoplevic resection (LEER) or a laterally extended parametrectomy (LEP), often require a dissection of the pelvic sidewall (PSW) [3,4,5,6,7,8,9]. Extraperitoneal internal iliac artery ligation can also be necessary for patients with locally advanced cervical cancer and severe genital bleeding. Anatomical variations in that region are often encountered. Therefore, in order to perform a dissection safely, gynecologists should be familiar with the anatomy of the PSW and possible anatomical variations in it. Dissection should be meticulous, and detailed knowledge of anatomy is mandatory. Moreover, there are many controversies at that region among oncogynecologists [10,11,12]. In particular, several imprecise or confusing definitions exist in regard to the region located medially from the psoas muscle. Therefore, after discussing the anatomy of the PSW and the commonly used anatomical terminology, we define a new term and boundaries of a potential avascular space, the medial psoas space. Contrary to the variety of earlier definitions, the proposed boundaries relate to a truly avascular space and could help surgeons avoid complications resulting from misleading anatomical descriptions. The aims of the present study were: an accurate description of the PSW anatomy suitable for the gyneco-oncological approach; a clear delineation of a useful avascular space, the medial psoas space (MPS), which has been inconsistently defined in the literature; and the standardization of the PSW dissection technique based on an accurate anatomical description.

## 2. Materials and Methods

### Surgical Procedures

The description and reevaluation of the surgical anatomy, as well as the standardization of the surgical technique, were based on the retrospective analysis of 25 surgical procedures performed at the Department of Gynecology, University Hospital “Saint Anna” Varna. MPS and PSW dissections were performed in 25 gynecological cancer patients between February 2016 and December 2019. Sixteen patients had stage I and stage II cervical cancer. A lateral extended parametrectomy (LER) with selective pelvic lymphadenectomy was performed in 11 of them. The other five patients underwent pelvic lymphadenectomy (level one) followed by radical hysterectomy. For the remaining nine patients, six underwent pelvic lymphadenectomy followed by total abdominal hysterectomy, three of them for non-endometrioid endometrial cancer, and the other half for pelvic bulky lymph nodes due to ovarian cancer. The MPS was also accessed for one patient with undifferentiated uterine sarcoma adherent to the PSW. Extraperitoneal open internal iliac artery ligation after MPS dissection was performed in two patients with locally advanced cervical cancer and severe vaginal bleeding.

We performed the following surgical procedures and separated the PSW during surgery into three regions (compartments), consisting of the avascular compartment, vascular compartment and nerve compartment (Figure 1). The access to the PSW could be performed via an extra/transperitoneal approach and by open or minimally invasive surgery. The PSW and MPS dissection through the medial psoas approach by open surgery is described in the present article.

The anatomical limits of the PSW region are: Superiorly: the peritoneum covering the external iliac vessels and psoas major muscle.Inferiorly: the pelvic floor muscles, namely the piriformis, coccygeus, and iliococcygeus muscles.Ventrally: inguinal ligament.Dorsally: sacrum.Laterally: superiorly by the medial aspect of the psoas major muscle and inferiorly by the obturator internus muscle.Medially: superiorly by the external iliac vessels and inferiorly by the sacral spine [4,11,12,13].

The limits of the PSW regions are shown in Figure 2. The three regions (compartments) of the PSW are shown in Figure 3.

The technique of PSW and MPS dissection is described in three steps.

#### Step 1. Avascular Region

A vertical midline skin incision is performed below the umbilicus to the symphysis pubis. The subcutaneous fat, rectus adbominis muscle aponeurosis, transversalis fascia and visceral peritoneum are dissected and cut, whilst the rectus abdominis muscles are preserved. Pelvic intraabdominal organs are exposed. The ipsilateral ovary and fallopian tube are retracted medially for better exposure of the ovarian vessels. The round ligament is the key for the initial dissection of the MPS. The ligament is cut and ligated close to the PSW and laterally to the external iliac vascular system [14]. The posterior leaf of the broad ligament is incised in a ventro–dorsal direction from the round ligament to the common iliac artery bifurcation. The incision should be lateral to the external iliac vessels and medial to the psoas major muscle. In slim patients, it is possible that a transperitoneal visualization of the genitofemoral nerve should be conducted in order to avoid injury. Otherwise, after peritoneal incision, the genitofemoral nerve should be identified and retracted laterally to the iliacus fascia covering the psoas major muscle. A meticulous dissection of the areolar connective tissue between the iliacus fascia covering the psoas major muscle and external iliac vessels is performed. Afterwards, the external iliac vessels are dissected from the medial aspect of the psoas major muscle and medially mobilized. At the inferomedial part of the psoas major muscle, the obturator nerve is identified. The inferiorlimit of dissection is the obturator nerve. The avascular region is delimited laterally by the psoas major muscle, medially by the external iliac vessels, superiorly by the peritoneum, and inferiorly by the obturator nerve. Limits of the avascular region are shown in Figure 4. The avascular region dissection during an open surgery is shown in Figure 5.

#### Step 2. Vascular Region

Access to the vascular regions is made by an initial avascular region dissection. Obturator vessels and the majority of the parietal and visceral branches of the internal iliac artery and vein, which lie below the obturator nerve, are visualized below the external iliac vessels. The origin and draining patterns of superior gluteal, inferior gluteal, and internal pudendal vessels are identified. The parietal fatty tissue and the lymph nodes overlying these vessels are dissected. The vascular regions are delimited superiorly by the obturator nerve, inferiorly by the sacral plexus, laterally by the obturator internus muscle, and medially by the internal iliac artery. The limits of vascular region are shown in Figure 6 and the majority of vessels in the region in Figure 7.

Development of the lateral paravesical step and the lateral pararectal space (Latzko’s space) as well as the identification of the ureter could be performed, though not a necessary step. However, controlling iatrogenic injury is better achieved after the development of more avascular spaces and the visualization of the ureter.

#### Step 3. Nerve Region

We gain access to the nerve region after the dissection of the vascular region. The internal iliac vessels and the majority of their branches are ligated and divided. The internal iliac artery is ligated and divided at the site of branching off from the common iliac artery. Thereafter, the superior gluteal, interior gluteal, pudendal arteries and additional parietal branches were isolated, ligated and divided. The ligation of the internal iliac venous system started from the peripheral to the central, as Nishikimi et al. noticed that ligation and division from the main central trunk leads to the congestion and swelling of the peripheral internal iliac veins. Central to peripheral ligation may cause ligature slippage or the splitting of the peripheral internal iliac vein [4]. The division of the internal iliac system provides access to the sacral plexus. After the dissection of the parietal pelvic fascia, the sacral roots are identified directly at their emergence out of the sacral foramina. The sacral plexus lies over the sacrum and piriformis muscle, which is located posterior to the internal iliac vessels and ureter. The sacral plexus is formed by the lumbosacral trunk and the first to fourth sacral ventral rami. The lumbosacral trunk contains the ventral rami of the fourth and fifth lumbar spinal nerves. It is identified at the medial part of the psoas major muscle inferolateral to the obturator nerve and continues caudally over the pelvic brim to join the first sacral ramus. The sciatic nerve, also called the ischiadic nerve, is formed by the union of the lumbosacral trunk and the first to third sacral ventral rami. Through the greater sciatic foramen, the sciatic nerve leaves the pelvis, inferior to the piriformis muscle wrapping behind the ischial spine (under the sacrospinous ligament). Via the medial psoas approach, we can identify the lumbosacral trunk and the first to third sacral ventral rami. However, care should be taken in regard to the superior gluteal artery, which lies posterosuperiorly between the lumbosacral trunk and the first ventral rami and leaves the pelvis through the greater sciatic foramen superior to the piriformis muscle, whereas the inferior gluteal vessels lie inferolaterally between the second and third spinal ventral rami and leaves the pelvis through the greater sciatic foramen inferior to the piriformis muscle at the posterior aspect of the pudendal vessels. Furthermore, after the resection of the coccygeus muscle, the pudendal nerve and pudendal vessels are identified at the medial side of the inferior part of obturator internus muscle, which passes through the pudendal canal (Alcock’s canal) that corresponds to the perineal region below the pelvic floor [10,11,13,15]. The nerve region is delimited superiorly by the vascular region, inferiorly by the piriformis muscle, ventrally by the ischial spine, dorsally by the sacrum, laterally by the greater sciatic notch, and medially by the sacral spine [13]. The anatomical limits and access to the nerve region are shown in Figure 8 and Figure 9. The anatomy of Alcock’s canal is shown in Figure 10.

## 3. Results

Avascular region: The avascular region and MPS were dissected in all 25 patients. Only the MPS was encountered in eight patients who underwent pelvic lymphadenectomy (level one), five patients with cervical cancer and three women with non-endometrioid endometrial cancer.

Vascular region: Further dissection of the vascular region was performed in 17 patients, 11 with cervical cancer that underwent LER, three patients with ovarian cancer and bulky lymph nodes, one patient with uterine undifferentiated uterine sarcoma, and two patients with internal iliac artery ligation.

Nerve region: After the dissection of the vascular region the nerve region was encountered, but further dissection was not performed in this study.

## 4. Discussion

### 4.1. Terminological Controversies

The term “spaces” is considered to be areas delimited by at least two independent fasciae, which are filled with areolar connective tissue. These spaces are developed by separating two independent fascia along their cleavage plane [16]. This definition of space is anatomical rather than surgical. However, not every avascular space described in gynecological surgery is delimited by two independent fascia. For instance, the Fourth space (Yabuki space) or Okabayashi pararectal and paravaginal spaces are not delimited by at least two fascias [17,18]. Therefore, we define the term “avascular surgical spaces” as avascular areas filled with areolar connective tissue and delimited by two anatomical structures of nerve, vessel or ligament, which could be used during surgery in order to avoid possible injuries [2]. Moreover, a better control of eventual vessel injury is feasible when more avascular spaces are developed.

In the avascular regions of the PSW, a new term and boundaries of a potential avascular surgical space could be defined by using the medial psoas approach. Many authors described this space by using different terms. Denis Querleu clarified it as “laterovascular space” [10]. Marc Possover and Marcelo Ceccaroni defined it as “lumbosacral space” [15,19]. The term “iliolumbar space” has also been used [12]. Other authors named that space “obturator fossa” or “obturator space” [12,20]. Different boundaries of this space have been described among authors. Additionally, the majority of oncogynecologists defined the pelvic floor, piriformis muscle or sacral plexus as the inferior limit [12,15,19]. However, if these are the inferior boundaries, it could not be defined as “avascular space” as the obturator vessels and branches of the internal iliac artery/vein (superior/inferior gluteal, pudendal) are located below the obturator nerve. Therefore, to define this space as “avascular”, the obturator nerve should be the inferior border of dissection. We called this space the “medial psoas space” and defined its boundaries (Figure 11 and Figure 12):Superior: peritoneum covering the psoas major muscle and external iliac vessels.Inferior: obturator nerve.Ventral: deep circumflex iliac vein.Dorsal: common iliac artery bifurcation.Lateral: fascia iliacus covering the psoas major muscle, genitofemoral nerve.Medial: external iliac artery and vein.

The MPS is a surgical avascular space, as only the lateral border is covered with fascia, iliacus fascia. Fascia covering the obturator internus muscle are not a part of the MPS as the muscle is located just below the obturator nerve [13].

**Figure 11 diagnostics-12-00519-f011:**
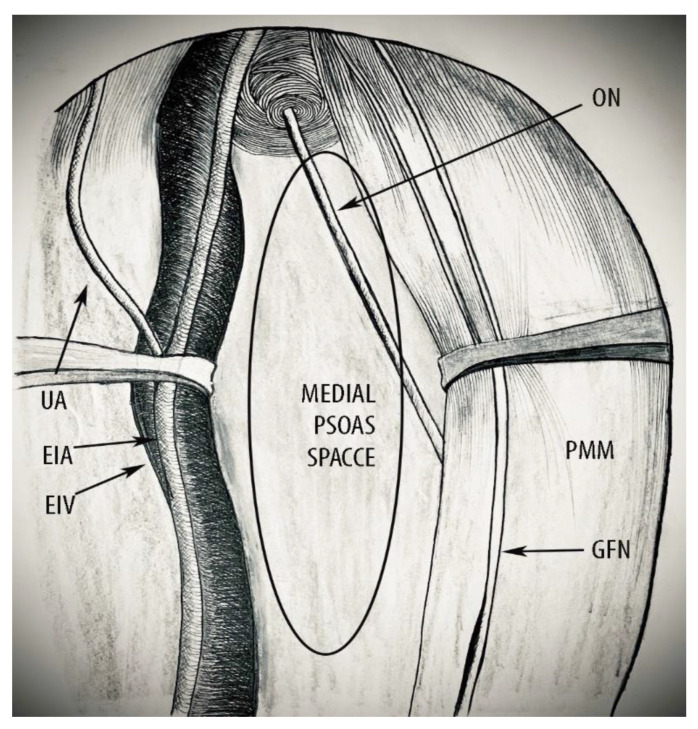
Boundaries of the medial psoas space (right pelvic sidewall). ON—obturator nerve; PMM—psoas major muscle; GFN—genitofemoral nerve; UA—umbilical artery; EIA—external iliac artery; EIV—external iliac vein; and UA—umbilical artery.

**Figure 12 diagnostics-12-00519-f012:**
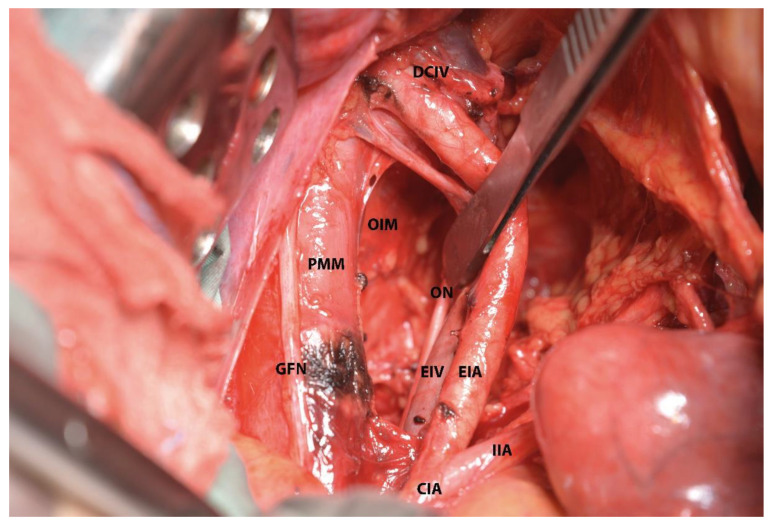
Medial psoas space (left pelvic sidewall). The inferiorlimit of the space—the obturator nerve is retracted medially. PMM—psoas major muscle; GFN—genitofemoral nerve; EIA—external iliac artery; EIV—external iliac vein; CIA—common iliac artery; IIA—internal iliac artery; ON—obturator nerve; DCIV—deep circumflex iliac vein; and OIM—obturator internus muscle.

### 4.2. MPS—Clinical Applications in Gynecologic Oncology

#### 4.2.1. MPS and Avascular Region

##### Pelvic Lymphadenectomy

The MPS could be dissected during pelvic lymphadenectomy for gynecological cancers. After encountering the MPS, the obturator nerve is identified. The obturator lymph nodes are dissected from the PSW and obturator nerve in a lateromedial direction. Then, the common iliac and external iliac lymph nodes are dissected in a caudocranial direction to the level of the deep circumflex artery. Finally, the lymph nodes located medial to the external iliac vessels and obturator nerve are dissected in a lateromedial direction to the medial border shaped by the internal iliac artery and obliterated umbilical artery; all pelvic lymph nodes are removed “en bloc”. This is level I pelvic lymphadenectomy according to the Cibula and Abu-Rustum classification [20]. According to the Querleu and Morrow classification, level I lymphadenectomy corresponded to external and internal iliac lymph node dissection [21]. In such cases of internal iliac lymph node dissection, the avascular region below the MPS should be dissected, which brings access to the cranial part of obturator lymph nodes and the interiliac region, which is inferior to the internal iliac artery and formed by the confluence of the common iliac vein. We believe that the medial psoas approach will provide a better dissection plane during pelvic lymphadenectomy, especially for the obturator lymph nodes.

In cases of bulky lymph nodes over the external iliac vessels extending to the psoas major muscle, the MPS could be dissected in order to identify the obturator nerve and provide a safe plane of dissection [22]. Here, the dissection should start from the level of the common iliac artery; firstly, the common iliac artery should be medially dissected and mobilized. To avoid an injury to the common iliac vein, the anatomical localization must be known: on the right side the common iliac vein lies inferolateral to the common iliac artery, and on the left side the common iliac vein lies inferomedial to the common iliac artery.

Moreover, the MPS could be encountered in cases of positive external and obturator sentinel lymph nodes, as these lymph nodes are most commonly affected in cervical cancer patients with FIGO stages IB1 to IIB. The removal of SLN in the external iliac, interiliac, and obturator area enables the evaluation of more than 80% of all SLN in patients with cervical cancer [23,24].

##### Internal Iliac Artery Ligation

The medial psoas space will be dissected together with the lateral pararectal space (Latzko’s space) during therapeutic or prophylactic internal iliac artery ligation for severe vaginal bleeding. We developed both spaces during two cases of extraperitoneal open internal iliac artery ligation. The dissection of both spaces provides an adequate control for possible iatrogenic injury of the internal iliac vein. Controlling pelvic hemorrhage is easier and safer when all potential spaces near the surgical field are encountered before different types of oncogynecological procedures.

#### 4.2.2. MPS and Vascular Region Dissection

The MPS could be developed to gain access to the vascular and nerve regions of the PSW.

##### Pelvic Tumors Adhesive to the PSW

These cases are often considered as inoperable as a complete resection or optimal cytoreduction cannot be achieved. Moreover, such tumors may invade the internal iliac vessel system. However, en bloc resection of the tumors along with the entire internal iliac vessel system to achieve optimal cytoreduction is feasible if the tumor does not involve the lumbosacral plexus or proximal sciatic nerve [4,5,6]. Nishikimi et al. reported a study of 20 patients with locally advanced ovarian cancer adherent to the PSW. The authors performed the removal of the entire internal iliac vessel system and the complete resection of the tumor. They concluded that the procedure is feasible, safe, and efficient for an improved prognosis. The authors used the MPS as a step of the procedure [4]. We developed the MPS and the vascular region in a patient with undifferentiated uterine sarcoma adherent to the left PSW. The superior/inferior gluteal and obturator vessels were all ligated and divided with the left internal iliac artery, which was ligated 3 cm distal to its bifurcation. The tumor was also firmly adherent to the external iliac artery. It was removed after a precise and meticulous dissection. An external iliac artery resection and crossover ileofemoral bypass were not needed.

##### Total Mesometrial Resection, Laterally Extended Parametrectomy (LEP) and Laterally Extended Endopelvic Resection (LEER)

In 1941, Mibayashi introduced “super-radical hysterectomy” for locally advanced cervical cancers. The procedure included the extended dissection of the cardinal ligament and the removal of the entire internal iliac vessel system [25]. Since then, similar extended procedures for locally advanced cervical cancer have been reported. Höckel introduced an operative procedure (LEER) that aims to remove tumors involving the adjacent PSW muscle along with the internal iliac vessels for the local control of advanced and recurrent malignancies of the lower female genital tract [5,6,7]. Ungár and Pálfalvi reported a more radical surgical procedure (LEP) aiming to remove the entire parametrial tissue from the PSW [8,9]. LEP was indicated for patients with cervical cancer, stage IB with lymph node metastases and stage IIB. The aim of these procedures was to obtain a complete PSW resection and locoregional control of a locally advanced tumor. Furthermore, the MPS was dissected in all three operative techniques as the entire internal iliac vessel system was removed during these mentioned procedures [5,6,7,8,9,25]. The MPS gives better access to the gluteal, obturator, and pudendal vessels. Between February 2016 and August 2016, we performed 11 LER procedures where the development of the MPS was a necessary step. Currently, LEP indications are debatable as guidelines for cervical cancer recommends the abandonment of hysterectomy in cases of positive pelvic lymph nodes [8,9,26]. However, an LEP could be performed in selected cases in which the tumor involves the soft structures of the PSW during a pelvic exenteration in order to achieve lateral-free margins.

##### Pelvic Sidewall Recurrences

Conversely to LEP, the LEER procedure is still indicated in patients with cervical cancer (T4N0M0) and PSW recurrences. The tumor should not be larger than 5 cm and the recurrence-free interval should be longer than 5 months. Achieving lateral-free margins is feasible if the tumor does not involve the sacral plexus. For patients with PSW recurrences after adjuvant treatment, the development of the MPS and vascular region is the preferable approach. The dissection of the MPS gives better access to the PSW in cases of lateral tumor fixation to the obturator internus, coccygeus, or iliococcygeus muscles [5,6,7,26].

#### 4.2.3. MPS and Nerve Region Dissection

As mentioned above, the dissection of the nerve region was not performed during the surgeries reported in the present article. However, the nerve region could be observed for deep infiltrating endometriosis and pelvic nerve tumors (osteochondrosarcomas, immature teratomas and schwannomas) located on the sacral plexus, obturator, pudendal or sciatic nerve [15,18,27]. Possover reported that best exposure of the lumbosacral trunk is obtained by a lumbosacral space dissection [27]. The author used the medialpsoas laparoscopic approach, strictly following the inferior border of the psoas major muscle. This approach allows for the identification of the lumbosacral trunk as a white band running along the linea terminalis, about 1 cm below the obturator nerve. Furthermore, the sciatic nerve could be encountered after distal dissection of the lumbosacral trunk.

### 4.3. Anatomical Variations Related to the MPS

The iliolumbar vein (ILV) drains the venous blood from the iliac fossa, psoas major and iliacus muscle, and terminates into the common iliac vein. The ascending lumbar vein (ALV) joins in the anastomotic venous system, which is formed by the inferior vena cava and the superior vena cava. The caudal part of the ALV drains into the cephalic border of the common iliac vein [2,13]. Lolis et al. described the detailed anatomy of the draining variations of the ILV and ALV based on a significantly great number of specimens: ILV/ALV draining separately into the CIV; ILV/ALV draining into the common iliac vein as a common trunk; and ILV draining into the external/internal iliac venous system. Furthermore, authors separate the draining variations into two types and noticed that ILV draining variations (91%) are higher when compared to the ALV (34%) [2,28]. Although the MPS is widely used, the draining variations of these vessels have recently been discussed in gynecologic oncology [2]. The possibility of such variations as they cross the avascular space should be noted and could be easily injured during dissection of the space. Moreover, we analyzed the draining variations of 12 patients (24 pelvic sidewalls) during MPS dissection. Draining variations of the ALV and ILV were observed in 50% (six) of the patients. Two patients had bilateral variations (16.6%). The total percentage of variations among 24 pelvic sidewalls (PSWs) was 33.3% (eight PSWs), of which 16.6% (four PSWs) had only one vein draining into the external iliac vein (EIV), 8.3% (two PSWs) had two veins draining into the EIV, 4.1% (one PSW) had three veins draining into the EIV, and 4.1% (one PSW) had two veins draining into the EIV via a common trunk (Figure 13). Although our research included a small group of patients, the importance of such variations during dissection should be stressed, as half of the patients had at least one draining variation of the ILV and ALV. Unfortunately, we could only speculate which of the veins are the ILV and ALV, as additional dissection in our surgeries was not necessary and could have been dangerous for the patients.

Oncogynecologists should also be aware of the variations in the external and internal iliac system [2,14].

In rare cases, an additional muscular branch from the external iliac artery descending laterally to the artery could be observed [29].

**Figure 13 diagnostics-12-00519-f013:**
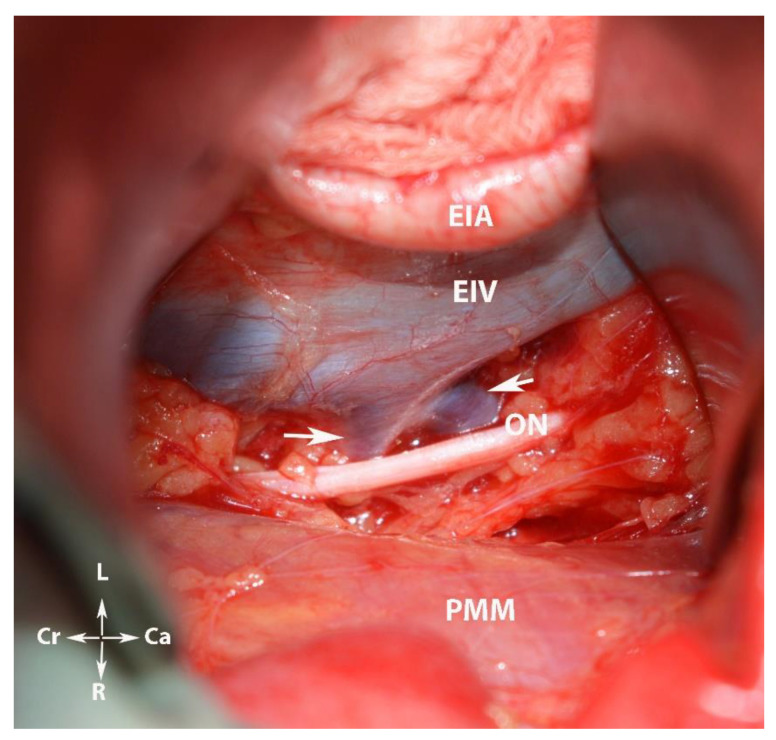
The ALV or ILV draining into the EIV as a common trunk (white arrows). EIA—external iliac artery; EIV—external iliac vein; ON—obturator nerve; PMM—psoas major muscle; Cr—cranial; Ca—caudal; L—left; and R—right.

## 5. Conclusions

Knowledge of the PSW anatomy is of the utmost importance during some oncogynecological procedures. The division of the PSW into three regions will better delineate the anatomy and technique of surgical procedures in that region. Emphasizing new insights concerning a potential avascular space may decrease morbidity during oncogynecological procedures in the PSW. The space gives better access to the obturator fossa and anatomical structures located below the obturator nerve in cases of metastatic lymph nodes or tumors to the PSW. Describing the clear boundaries of and possible anatomical variations in the MPS may urge oncogynecologists to consider different approaches during surgery.

## Figures and Tables

**Figure 1 diagnostics-12-00519-f001:**
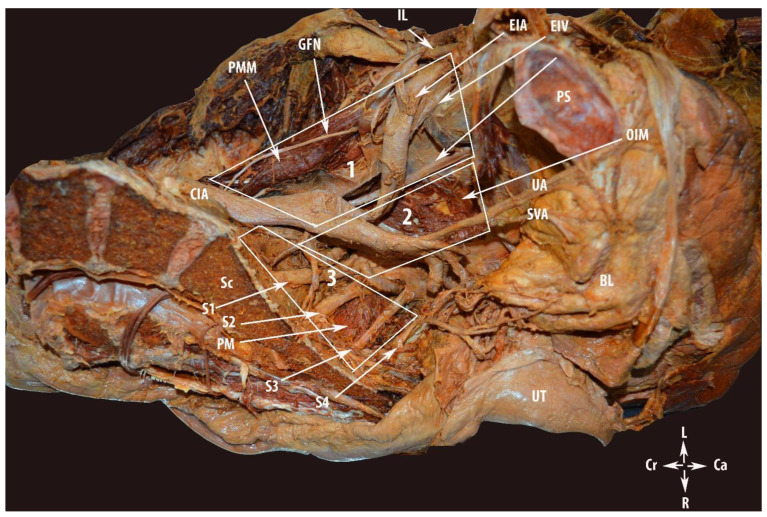
Regions of the pelvic sidewall in gynecology: 1—avascular region; 2—vascular region; and 3—nerve region. PMM—psoas major muscle; GFN—genitofemoral nerve; IL—inguinal ligament; CIA—common iliac artery; EIA—external iliac artery; EIV—external iliac vein; OIM—obturator internus muscle; UA—umbilical artery; SVA—superior vesical artery; PM—piriformis muscle; PS—symphysis pubis; UT—uterus; Sc—sacrum; BL—bladder; S1, S2, S3 and S4—anterior rami of the sacral spinal nerves; Cr—cranial; Ca—caudal; L—left; and R—right.

**Figure 2 diagnostics-12-00519-f002:**
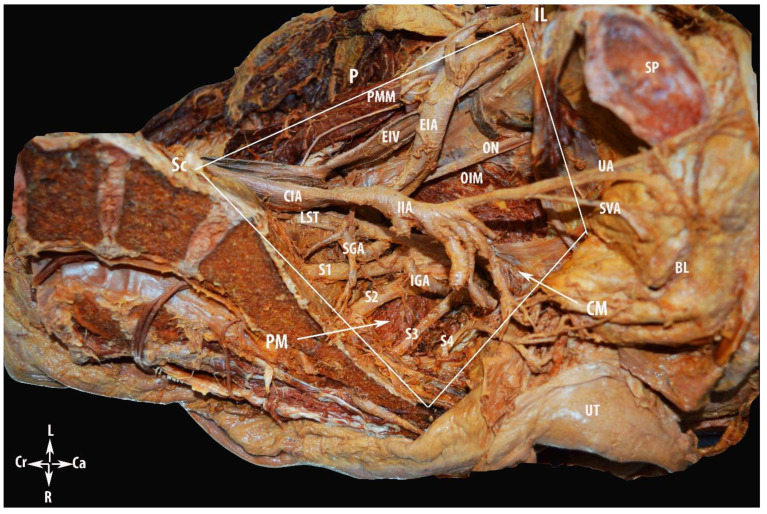
Limits of the pelvic sidewall region. IL—inguinal ligament; P—the peritoneum covering the external iliac vessels and psoas major muscle; Sc—sacrum; PMM—psoas major muscle; EIA—external iliac artery; IIA—internal iliac artery; EIV—external iliac vein; CIA—common iliac artery; ON—obturator nerve; OIM—obturator internus muscle; UA—umbilical artery; SVA—superior vesical artery; SGA—superior gluteal artery; IGA—inferior gluteal artery; SP—symphysis pubis; BL—bladder; UT—uterus; CM—coccygeus muscle; PM—piriformis muscle; LST—lumbosacral trunk; S1, S2, S3 and S4—anterior rami of the sacral spinal nerves; L—left; R—right; Cr—cranial; and Ca—caudal.

**Figure 3 diagnostics-12-00519-f003:**
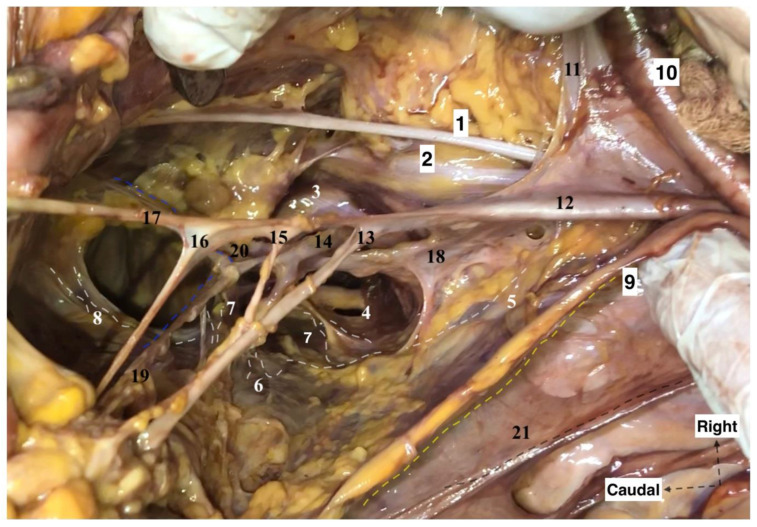
Right pelvic sidewall anatomy regarding the avascular, vascular and nerve compartments (cadaveric dissection by Dr. Ilker Selcuk). 1. Right obturator nerve, 2. Lumbosacral trunk, 3. S1 nerve, 4. S2 nerve, 5. Hypogastric nerve, 6. Inferior hypogastric plexus, 7. Pelvic splanchnic nerve bundles, 8. Radiating fibers of inferior hypogastric plexus under the terminal ureter at the paracolpium, 9. Ureter, 10. External iliac artery, 11. External iliac vein, 12. Internal iliac artery, 13. Uterine artery, 14. Pudendal artery, 15. Superior vesical artery, 16. Inferior vesical artery, 17.Obliterated umbilical artery, 18. Internal iliac vein, 19. Deep uterine vein, 20. Obturator vein, 21. Cut end of broad ligament—posterior leaf.

**Figure 4 diagnostics-12-00519-f004:**
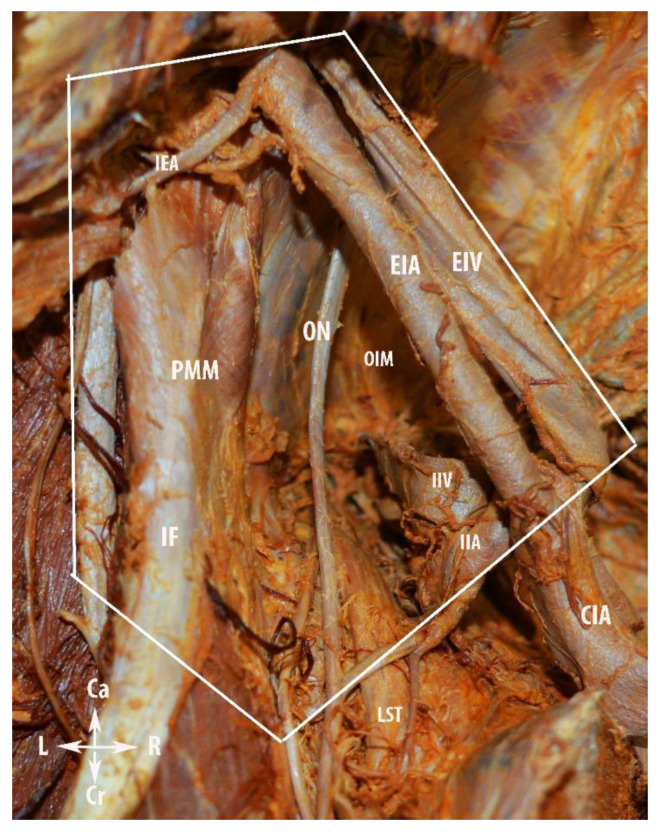
Limits of the avascular region of the pelvic sidewall. IEA—inferior epigastric artery; PMM—psoas major muscle; ON—obturator nerve; OIM—obturator internus muscle, LST—lumbosacral trunk; CIA—common iliac artery; IIA—internal iliac artery; IIV—internal iliac vein; EIA—external iliac artery; EIV—external iliac vein; IF—iliacus fascia covering the psoas major muscle; Ca—caudal; Cr—cranial; R—right; and L—left.

**Figure 5 diagnostics-12-00519-f005:**
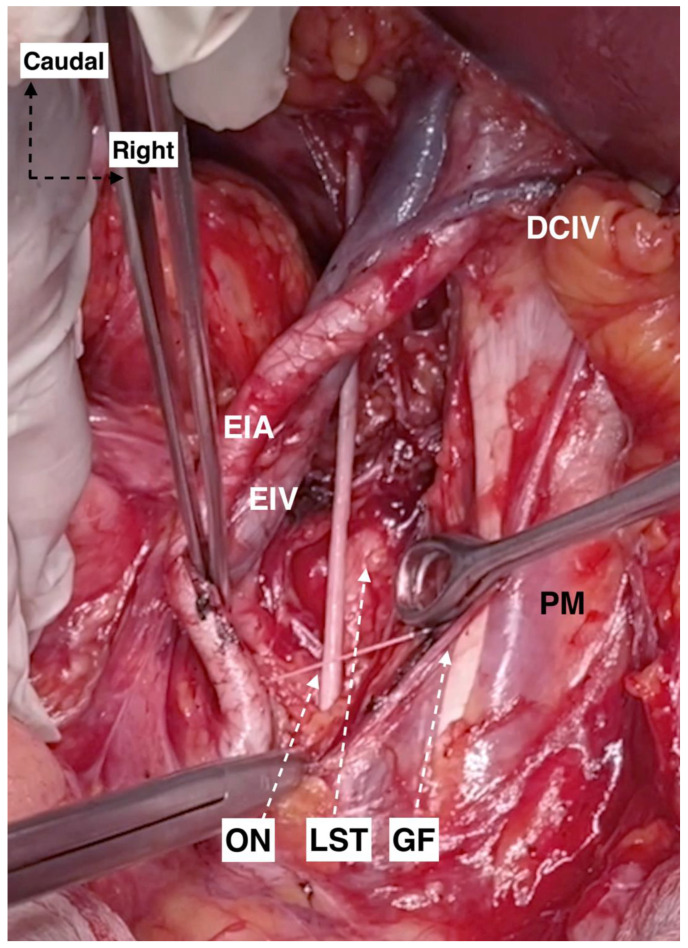
Avascular region of the right pelvic sidewall (surgical archive by Dr. Ilker Selcuk). DCIV: deep circumflex iliac vein; EIA: external iliac artery; EIV: external iliac vein; ON: obturator nerve; LST: lumbosacral trunk; GF: genitofemoral nerve; and PM: psoas major muscle.

**Figure 6 diagnostics-12-00519-f006:**
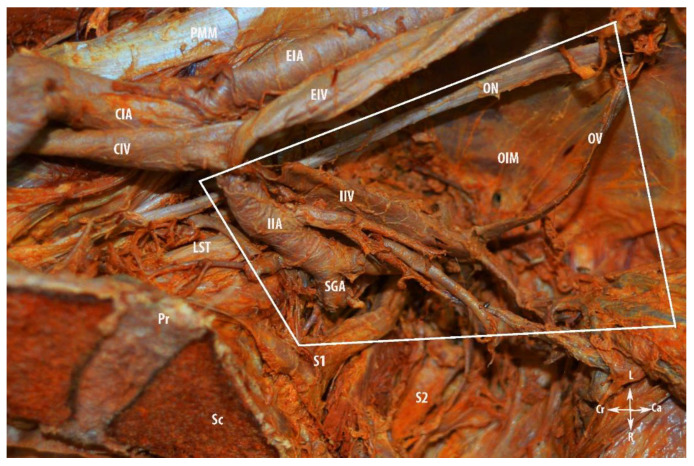
Limits of the vascular region of the pelvic sidewall. PMM—psoas major muscle; CIA—common iliac artery; CIV—common iliac vein; EIA—external iliac artery; EIV—external iliac vein; IIA—internal iliac artery; IIV—internal iliac vein; SGA—superior gluteal artery; OV—obturator vein; OIM—obturator internus muscle; LST—lumbosacral trunk; Pr—promontory; Sc—sacrum; S1 and S2—anterior rami of the sacral spinal nerves; Ca—caudal; Cr—cranial; L—left; R—right.

**Figure 7 diagnostics-12-00519-f007:**
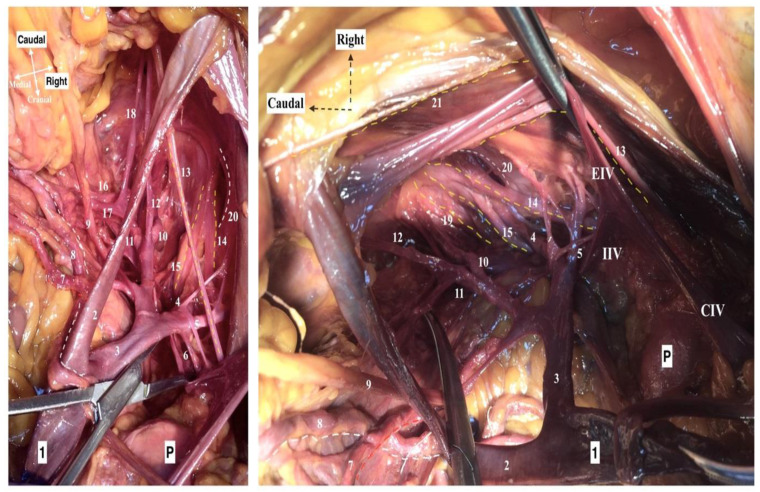
Vascular region of the right pelvic sidewall (cadaveric dissection by Dr. Ilker Selcuk). 1. Common iliac artery, 2. External iliac artery, 3. Internal iliac artery, 4. Superior gluteal artery, 5. Iliolumbar artery, 6. Lateral sacral artery, 7. Uterine artery (Red line), 8. Ureter (White line), 9. Umbilical artery (Obliterated), 10. Inferior gluteal artery, 11. Internal pudendal artery, 12. Obturator artery, 13. Obturator nerve (Yellow line), 14. Lumbosacral trunk (Yellow line), 15. S1 Nerve (Yellow line), 16. Middle rectal artery, 17. Deep uterine vein, 18. Obturator vein, 19. S2 Nerve (Yellow line), 20. Greater sciatic foramen, 21. Genitofemoral nerve. P—promontory, CIV—common iliac vein, EIV—external iliac vein, IIV—internal iliac vein, IIA—internal iliac artery.

**Figure 8 diagnostics-12-00519-f008:**
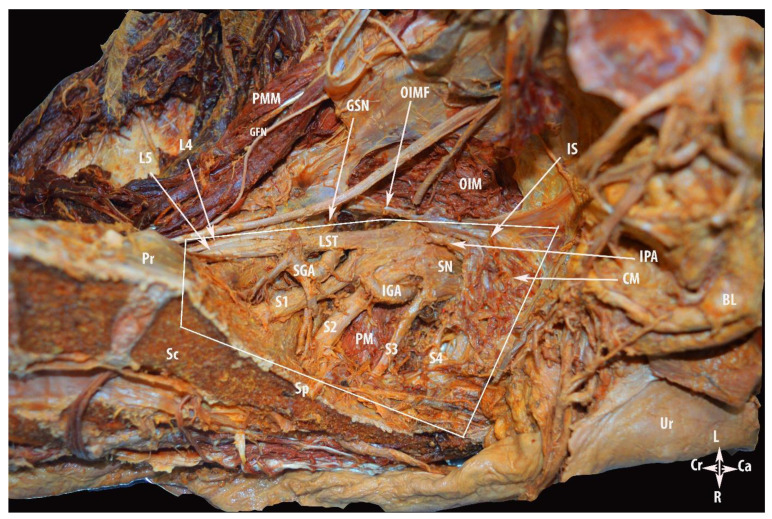
Limits of the nerve region of the pelvic sidewall. For better visualization of anatomical structures, the external iliac vessels and part of the internal iliac vessels are removed. PMM—psoas major muscle; GFN—genitofemoral nerve; GSN—greater sciatic notch; OIMF—obturator internus muscle fascia; OIM—obturator internus muscle; LST—lumbosacral trunk; S1, S2, S3 and S4—anterior rami of the sacral spinal nerves; L4 and L5—anterior rami of the lumbar spinal nerves; SGA—superior gluteal artery; IGA—inferior gluteal artery; IPA—internal pudendal artery; PM—piriformis muscle; SN—sciatic nerve; CM—coccygeus muscle; IS—ischial spine; BL—bladder; Ur—uterus; Sp—sacral spine; Sc—sacrum; Pr—promontory; L—left; R—right; Ca—caudal; and Cr—cranial.

**Figure 9 diagnostics-12-00519-f009:**
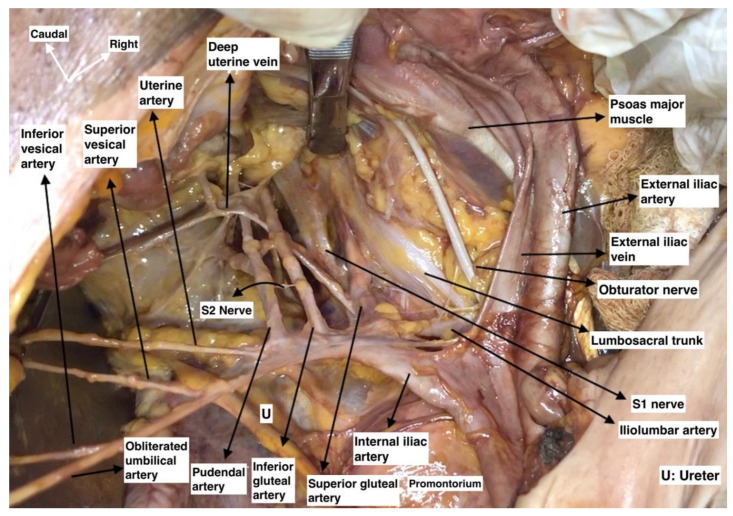
Access to the nerve region at the right pelvic sidewall (cadaveric dissection by Dr. Ilker Selcuk).

**Figure 10 diagnostics-12-00519-f010:**
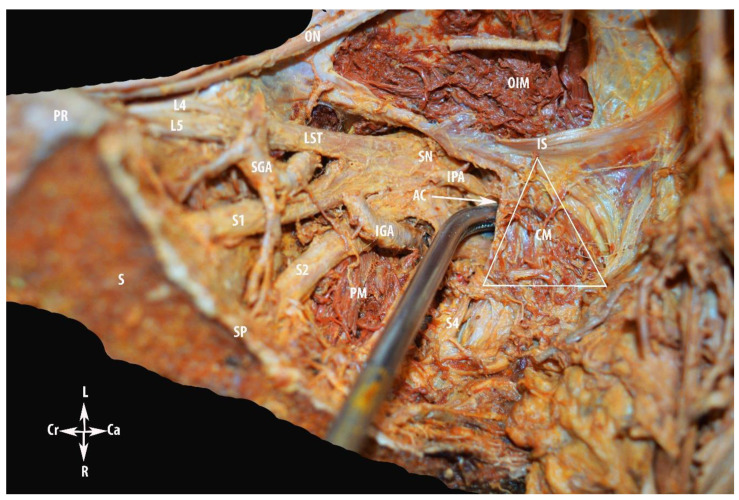
Alcock’s (pudendal) canal. ON—obturator nerve; OIM—obturator internus muscle; CM—coccygeus muscle; PM—piriformis muscle; SGA—superior gluteal artery; IGA—interior gluteal artery; IPA—internal pudendal artery; L4 and L5—anterior rami of the lumbar spinal nerves; LST—lumbosacral trunk; S1 and S2—anterior rami of the sacral spinal nerves; SN—sciatic nerve; IS—ischial spine; AC—entry to Alcock’s canal; S—sacrum; PR—promontory; SP—sacral spine; L—left; R—right; Ca—caudal; and Cr—cranial.

## Data Availability

The authors declare that all related data are available concerning researchers by the corresponding author’s email.

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
