# Peer review of "Pelvic Sidewall Anatomy in Gynecologic Oncology—New Insights into a Potential Avascular Space"

_diagnostics, 2022, doi:10.3390/diagnostics12020519_

Round 1

Reviewer 1 Report

I read with great interest this work. 

I am truly impressed by the quality of the anatomical descriptions and the pictures of cadavers dissections. This work could contribute to anatomical formation during surgical courses. It is well written and easily understandable.

The only remark is to uniform the legends on the figures (including the style and police)

Congratulation

Author Response

Reviewer 1.

I read with great interest this work.

I am truly impressed by the quality of the anatomical descriptions and the pictures of cadavers dissections. This work could contribute to anatomical formation during surgical courses. It is well written and easily understandable.

The only remark is to uniform the legends on the figures (including the style and police)

Congratulation

  1. The only remark is to uniform the legends on the figures (including the style and police)

Author’s Reply:

 We completely agree with the reviewer. The legends on the figures have been uniformed.

We are grateful for your valuable time and effort in reviewing our manuscript.

Based on your useful and scientific comments, we believe our manuscript has been

improved to a higher level.

Reviewer 2 Report

I think that this manuscript would be more appropriate for a surgery or gynecology journal. Moreover, the authors should recognize that this space has already been described in several articles:

Ceccaroni et al. 2010, DOI 10.1007/s00276-010-0624-6

Kale et al. 2018, doi.org/10.1177/2284026518798331

Cosma et al. 2021, doi.org/10.1007/s12565-020-00553-z

Possover et al., 2021, doi.org/10.1016/j.jmig.2021.12.004

Author Response

I think that this manuscript would be more appropriate for a surgery or gynecology journal. Moreover, the authors should recognize that this space has already been described in several articles:

Ceccaroni et al. 2010, DOI 10.1007/s00276-010-0624-6

Kale et al. 2018, doi.org/10.1177/2284026518798331

Cosma et al. 2021, doi.org/10.1007/s12565-020-00553-z

Possover et al., 2021, doi.org/10.1016/j.jmig.2021.12.004

  1. I think that this manuscript would be more appropriate for a surgery or gynecology journal.

Author’s Reply:

We do not agree with the reviewer that the manuscript is more appropriate for a surgical or gynecological journal. Most of the figures are from cadavers, not from surgeries. Furthermore, one of the keywords in the special issue is “ Surgical anatomy”. We believe that anatomical description and their applications in gynecologic oncology in the manuscript are associated with the term “ Surgical anatomy”. Moreover, the current recent advances in anatomy are thanks to the close relationship between anatomy and its clinical implication in different types of surgery. The majority of articles in the special issue are clinically oriented.

  1. Moreover, the authors should recognize that this space has already been described in several articles:

Author’s Reply: We do not agree with the reviewer.

We are familiar with all the articles, especially with the first and fourth articles as the authors were cited in the manuscript. The understanding of authors about the avascular space is the same in the majority of their articles. Additionally, as it was mentioned in the manuscript, authors described an avascular space in the pelvic sidewall, but they used different terminology. Furthermore, the boundaries do not corresponded to a potential avascular space (that was clearly explained in the manuscript). According to the reviewer, this space has already been described in several articles. We believe that this is not correct and irrelevant, as different terminology is used. Moreover, in order to describe an avascular space one should discuss, delineate and emphasize its all boundaries and to define if this space fulfill the criteria for surgical avascular space. In none of the mentioned articles by the reviewer, the boundaries and criteria for surgical avascular space have been described and discussed.  It is logical, as although all articles are amazing their clinical purpose, topic and significance completely differ from the concept, questions and discussions of our manuscript.

Additionally, if one avascular space is described in the past, this does not mean that all details of that space are defined. As surgeons believe that development of avascular spaces is crucial, new articles clarifying and developing these spaces are often encountered in medical journals nowadays.  Articles of avascular spaces in the pelvis continue to rise controversies among surgeons. Yabuki and Okabayashi changed limits of theirs avascular spaces numerous times. Moreover, other authors have different understanding of some of the avascular spaces. Liang et al. concluded that the Yabuki space should be located between the lateral side of the vagina and the caudal portion of the vesico-cervical ligament.

Yabuki et al., 2000, doi.org/10.1006/gyno.1999.5723

Yabuki et al., 2004, doi: 10.1016/j.ajog.2005.02.108.

Liang et al., 2010, doi.org/10.1016/j.ygyno.2010.06.033

Yoshihiko Yabuki, 2020, doi.org/10.1016/j.gore.2020.100623

 Therefore, we believe that the concept and clinical applications of our manuscript is applicable.

  1. Ceccaroni et al. 2010, DOI 10.1007/s00276-010-0624-6

Author’s Reply:  The article was cited in the manuscript. The differences between author understanding and terminology of the avascular space in the pelvic sidewall and our understanding and explanations were clearly highlighted in the manuscript. Author used different terminology (lumbo-sacral space) and did not describe the limit of the avascular space.

  1. Kale et al. 2018, doi.org/10.1177/2284026518798331

Author’s Reply:  In the article authors also used different from our terminology – “Iliolumbar space”. We mentioned in the manuscript that some authors used such a terminology for that space. Nevertheless, authors did not explain the clear boundaries of that space.

Moreover, as it was stressed in our manuscript, the inferior border of medial psoas space in the obturator nerve, because below the nerve are the branches of the internal iliac artery and it could not be define as “Avascular”.

Kale et al. described that the lumbosacral trunk and the distal portions of the S1, S2, S3 and S4 nerve roots merge in this space. Therefore, it could not be define as avascular space because the vascular region is above the nerve region and surgeons should dissect the vascular region first.

  1. Cosma et al. 2021, doi.org/10.1007/s12565-020-00553-z

Author’s Reply: We are familiar with Cosma et al. article as it was cited in one of our new articles.  Authors divided the pelvic retroperitoneum into four compartments: parietal, laterally to the umbilicovesical fascia; vascular, between the two fasciae; neural, medial to the urogenital-hypogastric fascia and visceral between the sacropubic ligaments.

However, Cosma et al. did not discussed entirely the pelvic sidewall. Moreover, authors defined the majority of pelvic spaces, but the medial psoas space or any space located between the psoas major muscle and external iliac vessels, together with its boundaries is not mentioned in the article.

  1. Possover et al., 2021, doi.org/10.1016/j.jmig.2021.12.004

Author’s Reply: The same as Ceccaroni, author used different terminology and boundaries of the space are not clearly defined. Moreover, the video article described the LION procedure, which is not the topic in the present manuscript

We are grateful for your valuable time and effort in reviewing our manuscript.

Reviewer 3 Report

I think the manuscript is interesting and very "captivating" for those involved in pelvic surgery.
The concept of medial psoas space is very intriguing, as a potential avascular space, to be explored in a greater number of interventions.
The anatomical details exhibited, together with the description of the surgical landmarks, is complete and accurate. The images are exhaustive.
I think that, after a revision of the English language, it can be accepted.

Author Response

I think the manuscript is interesting and very "captivating" for those involved in pelvic surgery.

The concept of medial psoas space is very intriguing, as a potential avascular space, to be explored in a greater number of interventions.

The anatomical details exhibited, together with the description of the surgical landmarks, is complete and accurate. The images are exhaustive.

I think that, after a revision of the English language, it can be accepted

  1. I think that, after a revision of the English language, it can be accepted.

Author’s Reply: The English MPDI editing services has been used in order to improve the English grammar  in the manuscript.

We are grateful for your valuable time and effort in reviewing our manuscript.

Based on your useful and scientific comments, we believe our manuscript has been improved to a higher level.

Round 2

Reviewer 2 Report

1. The authors recognize that the space has already been described. Therefore, they ought to modify the title and the conclusion.

2. The aim of the study should be stated in the introduction.

3. The authors state: ‘MPS and PSW dissections were performed on25 gynecological cancer patients between February 2016 and December 2019’. Where were these dissections performed? The list of the authors includes authors who come from four different Universities.

4. The date and the number of Ethics Committee permission are missing.

Author Response

Reviewer comments:
1. The authors recognize that the space has already been described. Therefore, they ought to
modify the title and the conclusion.
Author’s Reply: We agree with the reviewer. Although different definitions and not clear
boundaries have been described, some authors recognized that space. We used the term “a
new avascular space” as in our opinion the different author’s definitions did not corresponded
to a real avascular space. However, some limits and access of the space were described in the
past (although not accurately). Therefore, we change the title and conclusion.
New title was inserted:
“Pelvic sidewall anatomy in Gynecologic Oncology – New Insights into a Potential Avascular
space”.

The conclusion was also changed.

2. The aim of the study should be stated in the introduction.
Author’s Reply: We agree with the reviewer.

The next text was inserted:
The aims of the present study were: an accurate description of the PSW anatomy suitable for
the gyneco-oncological approach; a clear delineation of a useful avascular space, the medial
psoas space (MPS), which has been inconsistently defined in the literature; and the
standardization of the PSW dissection technique based on an accurate anatomical
description.”

3. The authors state: ‘MPS and PSW dissections were performed on25 gynecological cancer
patients between February 2016 and December 2019’. Where were these dissections
performed? The list of the authors includes authors who come from four different
Universities.
.
Author’s Reply: We agree with the reviewer. The primary investigation and surgical
procedures were perfomred in our hospital “ Saint Anna”. However, we collaborated with
anatomists and oncogynecologists in order to improve the interpretation of the surgical
anatomy findings. By that way, we used cadaveric and surgical photos of doctor Selcuk work
(an anatomists and oncogynecologists) in order to define the zones in the PSW more
appropriately with correct anatomical landmarks.
The text was inserted:
The description and reevaluation of the surgical anatomy, as well as the standardization of
the surgical technique were based on the retrospective analysis of 25 surgical procedures
performed at the Department of Gynecology, University Hospital “Saint Anna” Varna.
4. The date and the number of Ethics Committee permission are missing.

Author’s Reply: The data and number of Ethics Committee permission were inserted in the
manuscript.
We are grateful for your valuable time and effort in reviewing our manuscript.
Based on your useful and scientific comments, we believe our manuscript has been
improved to a higher level.

Round 3

Reviewer 2 Report

All the comments were addressed and the manuscript can now be accepted.